

# Healthcare applications of single camera markerless motion capture: a scoping review

Bradley Scott[1], Martin Seyres[2], Fraser Philp[3], Edward K. Chadwick[2] and Dimitra Blana[1]

[1] School of Medicine, Medical Sciences and Nutrition, University of Aberdeen, Aberdeen, United Kingdom
[2] School of Engineering, University of Aberdeen, Aberdeen, United Kingdom
[3] School of Health Sciences, University of Liverpool, Liverpool, United Kingdom

Corresponding author
Bradley Scott, b.scott.20@abdn.ac.uk

## ABSTRACT

**Background.** Single camera markerless motion capture has the potential to facilitate at home movement assessment due to the ease of setup, portability, and affordable cost of the technology. However, it is not clear what the current healthcare applications of single camera markerless motion capture are and what information is being collected that may be used to inform clinical decision making. This review aims to map the available literature to highlight potential use cases and identify the limitations of the technology for clinicians and researchers interested in the collection of movement data.

**Survey Methodology.** Studies were collected up to 14 January 2022 using Pubmed, CINAHL and SPORTDiscus using a systematic search. Data recorded included the description of the markerless system, clinical outcome measures, and biomechanical data mapped to the International Classification of Functioning, Disability and Health Framework (ICF). Studies were grouped by patient population.

**Results.** A total of 50 studies were included for data collection. Use cases for single camera markerless motion capture technology were identified for Neurological Injury in Children and Adults; Hereditary/Genetic Neuromuscular Disorders; Frailty; and Orthopaedic or Musculoskeletal groups. Single camera markerless systems were found to perform well in studies involving single plane measurements, such as in the analysis of infant general movements or spatiotemporal parameters of gait, when evaluated against 3D marker-based systems and a variety of clinical outcome measures. However, they were less capable than marker-based systems in studies requiring the tracking of detailed 3D kinematics or fine movements such as finger tracking.

**Conclusions.** Single camera markerless motion capture offers great potential for extending the scope of movement analysis outside of laboratory settings in a practical way, but currently suffers from a lack of accuracy where detailed 3D kinematics are required for clinical decision making. Future work should therefore focus on improving tracking accuracy of movements that are out of plane relative to the camera orientation or affected by occlusion, such as supination and pronation of the forearm.

## INTRODUCTION

Three-dimensional motion capture using multiple camera systems has been used to inform biomechanical analysis for several decades now (*Topka, Konczak & Dichgans, 1998*; *Corazza et al., 2006*; *Van den Noort et al., 2016*), allowing the quantification and identification of complex movement patterns and impairments. The biomechanical information can be used to inform clinical decision making, provided there is a clear link to an appropriate intervention *e.g.*, surgery for hamstring lengthening in children with cerebral palsy (*Arnold et al., 2006*; *Laracca et al., 2014*). Systems used in clinical analysis are predominantly marker-based; that is, they rely on tracking the trajectories of reflective markers attached directly to the body of the person under investigation. They are reliant on the availability of substantial infrastructure such as well-defined and validated marker sets and models, multiple cameras in a large capture volume, complex processes for capturing and processing data and technically experienced laboratory staff. Despite the ability of this technology to provide further information and the identified benefits for patient outcomes as a result of improved decision making (*Salami et al., 2019*; *Osborne et al., 2019*), these systems are not widely used across multiple health conditions and tend to be focused on a few specific patient populations such as children with cerebral palsy.

Whilst significant demonstrable benefits, such as decreasing unnecessary surgery or reducing cost to the service or a patient, may be the main driver for the adoption of this technology, it is also likely that the logistical and operational challenges associated with using this technology limit its overall clinical applicability for some patient groups (*Philp et al., 2021*). Camera-based, 3D analysis is reliant on specialist technical skills and processes regarding capture, reporting and interpretation of data which are time consuming and resource intensive. Furthermore, services which provide 3D movement analysis are typically housed in specialist centres which cover large geographical areas.

Markerless motion capture has been growing in interest over recent years, and may offer advantages over the gold-standard 3D marker-based motion capture approach, but it is recognised that there's a trade off in some cases between usability and accuracy (*Kanko et al., 2021*). The benefit to patients is that it is less intrusive, as it eliminates marker setup, which requires palpation and removal of clothing for exposure of bony anatomical landmarks. For the clinician, a reduction in the requirement to complete additional processes such as marker placement or manually input user data, *e.g.*, anthropometric measurements, more easily integrates these methods into the patient pathway (*Jaspers et al., 2011*). However, state-of-the-art markerless systems with comparable accuracy to marker-based optoelectronic systems, such as Theia3D (Theia Markerless Inc., Kingston, ON, Canada), still require multiple cameras as well as complex software and calibration, and come at a high cost.

Single camera markerless motion capture systems (SCMoCap) offer increasingly promising alternatives as the camera technology improves. Here we refer to both specialised devices such as the Kinect that include pose-estimating software, and standard cameras that require separate software like OpenPose. SCMoCap systems have the potential to facilitate remote, at home, or community movement assessments by reducing restrictions around

setup with multiple cameras. For the patient this makes for a more comfortable assessment and reduces travel to the clinic, which may already be difficult because of limited mobility. For the clinician this allows for a more natural observation of movement and may facilitate more frequent assessment.

Integration of technology into care pathways is a priority for healthcare services and the importance of this has been highlighted by the COVID-19 pandemic (*Royal College of Physicians, 2020*; *UK Government, 2021*; *World Health Organization, 2021*). SCMoCap technology may be an important tool for helping to deliver digitally integrated healthcare services. However, it is not clear what the current healthcare applications of SCMoCap are and what information is being collected that may be used to inform clinical decision making. Recent improvements to portable and affordable motion capture technology such as the Microsoft Kinect and Leap Motion, which include depth sensors in addition to conventional RGB video cameras, as well as the increased availability of high-resolution smartphone cameras, has inspired an increase in the investigation of SCMoCap for healthcare. Recent reviews of markerless motion capture have been conducted on gait (*Wade et al., 2022*) and 3D pose estimation (*Desmarais et al., 2021*). However, they focus mainly on the processing methods and accuracy evaluation of multi-camera motion capture systems with limited discussion on SCMoCap and clinical applications. The current state of research in this area thus welcomes a systematic approach to map the available literature to describe current practice. This review provides insight into the healthcare applications of SCMoCap to inform clinicians who are interested in implementing this technology into their practice, or researchers who are aiming to collect movement data with a single camera approach.

A scoping review was therefore conducted with the aims of investigating: which SCMoCap systems have been used for healthcare applications and for which target populations; what these systems measure; how their performance is evaluated; and the limitations of current systems and what must be improved so they are more widely used in the clinic.

## SURVEY METHODOLOGY

### Review protocol

This review was conducted and reported according to the PRISMA Extension for Scoping Reviews (PRISMA-ScR) (*Tricco et al., 2018*).

### Search strategy

A search was conducted in Pubmed, CINAHL and SPORTDiscus. Figure 1 summarises the stages of article identification, screening, eligibility, and inclusion.

Studies were collected up to 14 January 2022. The following medical Subject headings (Mesh) and key words were used: (application[Title/Abstract] OR software[MeSH] OR system[Title/Abstract] OR deeplabcut[Title/Abstract] OR openpose[Title/Abstract] OR neural network*[title/abstract]) AND (capture[Title/Abstract] OR camera[Title/Abstract] OR mls[Title/Abstract] OR kinect[Title/Abstract] OR 2d[Title/Abstract] OR 3d[Title/Abstract] OR rgb-d camera[Title/Abstract] OR pose estimation[Title/Abstract] OR movement[MeSH] OR motion[MeSH] OR

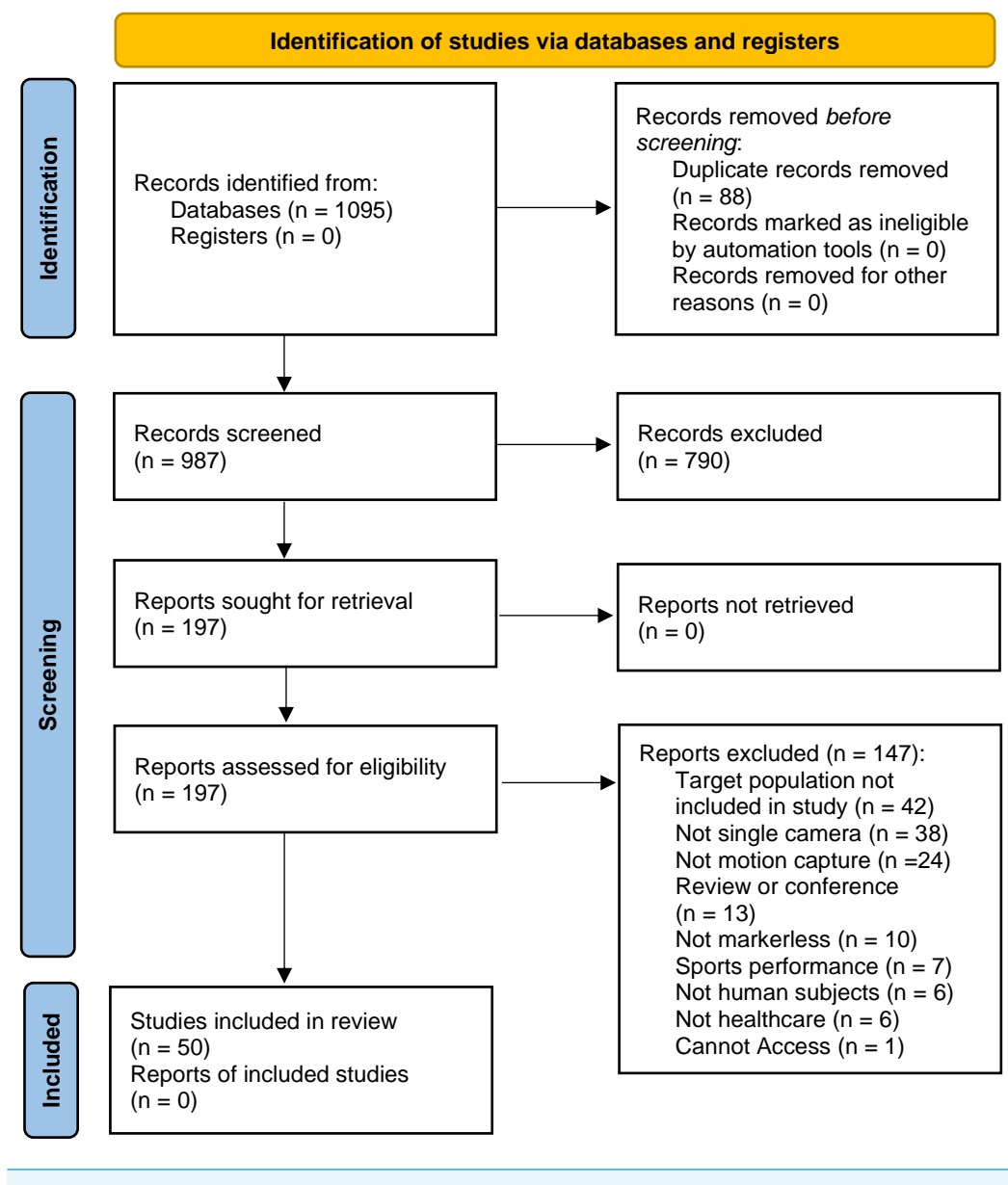

**Figure 1 PRISMA flow chart.**

marker*[Title/Abstract]) AND (markerless[Title/Abstract] OR marker-less[Title/Abstract] OR "single-camera"[Title/Abstract] OR "single camera"[Title/Abstract]).

## Inclusion criteria

Healthcare applications of SCMoCap was the focus of this study. Articles were included that involved the design, development, evaluation or use of SCMoCap systems for an explicitly identified patient group. All quantitative study designs were included in this review. Only studies published in English were considered.

### Exclusion criteria

Articles that focussed on animals, involved more than one camera, did not have an explicit stated use for healthcare or did not use markerless motion capture were excluded from this study. Articles that did not include the intended target population in the study were also excluded.

### Screening and article inclusion

Two authors (BS and MS) performed screening of title and abstract, and full text before data extraction. Disagreements in inclusion were resolved through discussion with a third author (FP).

### Data extraction

A total of 987 articles were identified after duplicate exclusion for the evaluation of title and abstract. A total of 197 articles were selected for full text examination. The final number of articles used in the data extraction was 50.

Table S1 details movement parameters collected in detail.

To address the scoping review aims, data was grouped according to the patient populations evaluated. A description of the single camera system, sample size and patient demographics was included. Any clinical outcome measures used and biomechanical data extracted from the SCMoCap systems was mapped according to The International Classification of Functioning, Disability and Health Framework (ICF) (*World Health Organization, 2001*). Additionally, any reported limitations, methods of evaluation and measurement errors were evaluated.

## RESULTS

### Patient group classifications

Patient groups were classified according to the following headings: Neurological Injury in Children; Neurological Injury in Adults; Hereditary/Genetic Neuromuscular Disorders; Frailty; and Orthopaedic or Musculoskeletal. A detailed description of study populations and groupings can be seen in Table 1.

### Neurological injuries in children

Neurological injuries in children *e.g.*, cerebral palsy, usually result from an injury to the brain and can occur prior to, during, or after birth, in the early years of life. They comprise a range of permanent sensory-movement disorders, and as a result, movement, balance and posture can be affected, with developmental delays noted in some cases (*Rosenbaum et al., 2007*; *Krägeloh-Mann & Cans, 2009*).

This is the only patient group out of the five identified in the search, for whom motion capture is routinely used. Typically consisting of 3D marker-based motion capture of gait combined with EMG recordings, the results are evaluated by a multi-disciplinary team, prior to deciding on a clinical intervention or decision such as surgery (*Arnold et al., 2006*; *Laracca et al., 2014*). This type of motion capture is typically applied to children of 5 years and above.

**Table 1** Target Populations.

| Population group | Population | Number of studies | Number of subjects range | Includes controls | Reference |
|---|---|---|---|---|---|
| Neurological Injuries in Children | | | | | |
| | Cerebral Palsy | 14 | 2–1,424 | Y | *Adde et al. (2010)*, *Stahl et al. (2012)*, *Chang, Han & Tsai (2013)*, *Rahmati et al. (2014)*, *Rahmati et al. (2016)*, *Orlandi et al. (2015)*, *Orlandi et al. (2018)*, *Støen et al. (2017)*, *Marchi, Hakala & Knight (2019)*, *Kidziński et al. (2020)*, *Schroeder, Hesse & Weinberger (2020)*, *Tsuji, Nakashima & Hayashi (2020)*, *Reich et al. (2021)* and *Groos et al. (2022)* |
| Neurological Injuries in Adults | | | | | |
| | Parkinson's | 9 | 5–119 | Y | *Galna et al. (2014)*, *Rocha et al. (2014)*, *Rocha et al. (2015)*, *Eltoukhy et al. (2017)*, *Butt et al. (2017)*, *Oña et al. (2018)*, *Grunert, Krause & Feig (2019)*, *Lee, Sinclair & Jones (2019)* and *Sato et al. (2019)* |
| | Stroke | 5 | 5–43 | Y | *Scano et al. (2014)*, *Vernon, Paterson & Bower (2015)*, *Ozturk et al. (2016)*, *Bonnechère et al. (2018)* and *Latorre et al. (2018)* |
| | Multiple Sclerosis | 3 | 44–149 | Y | *Behrens et al. (2014)*, *Behrens et al. (2016)* and *Grobelny et al. (2017)* |
| | Neurological Injury | 1 | 20 | N | *Lange et al. (2011)* |
| | Epilepsy | 2 | 17–111 | N | *Rémi et al. (2011)* and *Pereira Choupina et al. (2018)* |
| | Essential tremor (ET) | 1 | 4 | N | *Ozturk et al. (2016)* |
| | Cervical Dystonia | 1 | 30 | N | *Nakamura et al. (2019)* |
| | Dementia | 1 | 54 | N | *Mehdizadeh et al. (2021)* |
| | Chronic Disabilities Due to Neurological and Musculoskeletal Disorders | 1 | 57 | Y | *Capecci, Ceravolo & Ferracuti (2018)* |
| Hereditary/ Genetic Neuromuscular Disorders | | | | | |
| | Transthyretin Familial Amyloid Polyneuropathy (TTR-FAP) | 2 | 6–10 | N | *Carmo Vilas-Boas et al. (2019)* and *Vilas-Boas et al. (2019)* |
| | Hereditary Amyloidosis Associated with Transthyretin V30M (ATTRv V30M) | 1 | 66 | Y | *Vilas-Boas et al. (2020)* |
| | Facioscapulohumeral Muscular Dystrophy (FSHD) | 1 | 44 | Y | *Han et al. (2015)* |

*(continued on next page)*

**Table 1** (*continued*)

| Population group | Population | Number of studies | Number of subjects range | Includes controls | Reference |
|---|---|---|---|---|---|
| | Duchenne Muscular Dystrophy (DMD) | 1 | 53 | Y | *Han, Bie & Nicorici (2016)* |
| Frailty | | | | | |
| | Elderly | 3 | 7–17 | N | *Stone & Passive (2012)*, *Stone & Skubic (2013)* and *Stone & Skubic (2015)* |
| Orthopaedic or Musculoskeletal | | | | | |
| | Total Hip Replacement | 1 | 1 | N | *Dolatabadi, Taati & Mihailidis (2014)* |
| | Adhesive Capsulitis of the Shoulder (AC) | 1 | 27 | Y | *Lee, Yoon & Chung (2015)* |
| | Motor Disabilities Due to Different Unreported Pathologies | 1 | 33 | Y | *Capecci, Ceravolo & Ferracuti (2016)* |

Of the studies identified using SCMoCap, children with cerebral palsy were investigated in two studies: one involving gait (*Kidziński et al., 2020*) and one involving an upper limb rehabilitation game (*Chang, Han & Tsai, 2013*). *Kidziński et al. (2020)* used a regular video camera and a custom OpenPose workflow to track full body keypoints of the limbs, torso and head over time to calculate kinematic clinical outcome measures typically used in practice for treatment planning and monitoring. Outcomes included walking speed, cadence, joint angles, and predicted scores on Motor Function Classification (GMFCS) (*Palisano et al., 1997*) and Gait Deviation Index (GDI) (*Schwartz & Rozumalski, 2008*). Kinematics were found to have high correlation with ground truth marker data. GMFCS and GDI predictions were compared against clinical assessment and accuracy for these outcomes were reported to be close to the theoretical upper limit that exists due to patient stride variability (*Chia & Sangeux, 2017*).

In infant populations, the aim of movement analysis is to diagnose cerebral palsy by evaluating the absence or abnormality of "general movements", spontaneous non-goal-oriented movements that are displayed up to 20 weeks post-term, and may be an early indicator of cerebral palsy. These movements are clinically evaluated using General Movement Assessment (GMA). GMA has high sensitivity and specificity values (*Seesahai et al., 2021*), but may still misclassify patients as it is subjective and relies on user knowledge that is difficult to explain to others. As a result, the same movement can be interpreted differently by two observers. In an attempt to automate the assessment of infant movement, marker-based methods (*Meinecke et al., 2006*) and accelerometery (*Heinze et al., 2010*) have been used to evaluate general movements, but these methods require attachment of markers on the limbs, which can interrupt the natural occurrence of such movements (*Tsuji, Nakashima & Hayashi, 2020*).

SCMoCap offers a good opportunity to provide a similar automated assessment, but without the need to attach additional sensors to the infant, and a number of studies that attempted to do just this were identified in the literature. To assess general movements

in infants, movement trajectories, and velocities of the upper and lower limbs, which are the same kinematic features assessed through visual inspection by clinicians in the GMA, were measured. Most studies ($n = 9$) compared the results of their motion analysis with clinical GMA and found high agreement (*Adde et al., 2010*; *Orlandi et al., 2015*; *Orlandi et al., 2018*; *Støen et al., 2017*; *Marchi, Hakala & Knight, 2019*; *Schroeder, Hesse & Weinberger, 2020*; *Tsuji, Nakashima & Hayashi, 2020*; *Reich et al., 2021*; *Groos et al., 2022*). SCMoCap therefore has the potential to assess infant movement and assess gait measurements typically assessed in cerebral palsy populations without interference. Most of the studies identified ($n = 8$) (*Adde et al., 2010*; *Stahl et al., 2012*; *Rahmati et al., 2014*; *Rahmati et al., 2016*; *Orlandi et al., 2015*; *Orlandi et al., 2018*; *Støen et al., 2017*; *Kidziński et al., 2020*; *Tsuji, Nakashima & Hayashi, 2020*) used a 2D RGB camera to record movement. Using a 2D RGB camera limits the analysis to movements in a single plane, typically frontal for infant studies, but this was not a major limitation as measurements of axial rotation in this population are not a focus. Motion analysis carried out in 2D is also comparable to the approach taken in clinical practice where 2D video is used as part of GMA (*Aizawa et al., 2021*). *Kidziński et al. (2020)* demonstrated that accuracy loss that was not a result of patient stride variation could be reduced further by collecting kinematics with input from frontal and transverse planes. However, this is not a single camera solution and serves only to demonstrate the appropriateness of the machine learning model used to recognise movement. Another limitation reported in these studies involved the difficulty in mapping quantitative kinematic outputs from the motion capture measurements to more descriptive characteristics of motion assessed as part of a GMA, such as fluidity and elegance (*Orlandi et al., 2015*; *Tsuji, Nakashima & Hayashi, 2020*). This issue would also exist in marker-based systems as these concepts are not easy to objectively quantify with kinematics alone, but a SCMoCap approach offers an opportunity to conduct more research into these higher-level descriptors of movement by enabling more frequent data collection due to reduced setup and portability

## Neurological injuries in adults

Neurological injuries in adults can occur as a result of injury or damage to the brain or central nervous system. These can stem from several aetiologies *e.g.*, trauma (Traumatic brain injury), bleeds or blocks to the vascular system of the brain *e.g.*, stroke (*Lawrence et al., 2001*), autoimmune or immune mediated neural and glial damage *e.g.*, multiple sclerosis (*Isaksson, Ahlstrom & Gunnarsson, 2005*), or altered neuro-chemical processes *e.g.*, Parkinson's disease (*Dauer & Przedborski, 2003*) or epilepsy. Neurological injuries to the brain or central nervous system can affect several body systems such as visual or auditory, but in all cases of neurological injury, voluntary movement is affected.

In this group, SCMoCap was typically used to measure kinematics that occur during observer based ordinal or single measure clinical outcomes, in an effort to provide additional information that may be useful for treatment but is not typically assessed. The Microsoft Kinect v1, a hybrid RGB and infra-red depth sensing camera, was used in ten of the identified studies (*Lange et al., 2011*; *Behrens et al., 2014*; *Galna et al., 2014*; *Rocha et al., 2014*; *Vernon, Paterson & Bower, 2015*; *Behrens et al., 2016*; *Grobelny et al.,*

*2017*; *Bonnechère et al., 2018*; *Grunert, Krause & Feig, 2019*). Kinect v2, featuring a higher resolution, was used in six (*Rocha et al., 2015*; *Ozturk et al., 2016*; *Eltoukhy et al., 2017*; *Latorre et al., 2018*; *Pereira Choupina et al., 2018*; *Nakamura et al., 2019*) and the Leap Motion sensor, a smaller camera with similar technology, was used in four (*Chen et al., 2016*; *Grobelny et al., 2017*; *Butt et al., 2017*; *Oña et al., 2018*). Two studies used only an RGB camera (*Rémi et al., 2011*; *Sato et al., 2019*). Cameras with depth sensors as well as RGB sensors can record movements in 3D and have the potential to provide comparable accuracy to gold standard, 3D marker-based systems. This makes them particularly useful compared to 2D motion capture when dealing with motions where out of plane rotations are important.

In Parkinson's disease, physiotherapy has been demonstrated to increase functional ability (*Tomlinson et al., 2013*). In this population, inertial sensors (*Giggins, Sweeney & Caulfield, 2014*), tablet (*Dauvergne et al., 2018*), and mouse and keyboard (*Van de Weijer et al., 2019*) have been used to record motion for serious games for rehabilitation. However, these methods require a level of hand function that is typically only available to patients at an early stage of the disease, potentially excluding patients with more advanced Parkinson's from participating in the games. Inertial sensors may also be difficult to set up for home rehabilitation. A markerless approach can increase the effectiveness of rehabilitation by facilitating the continuous repetition of exercises at home. Most studies using SCMoCap for Parkinson's patients aimed to provide biofeedback for rehabilitation games (*Palacios-Navarro, García-Magariño & Ramos-Lorente, 2015*; *Capecci, Ceravolo & Ferracuti, 2018*; *Oña et al., 2018*; *Grunert, Krause & Feig, 2019*). *Oña et al. (2018)* reported measuring upper limb joint trajectories and range of motion with a Leap Motion controller to achieve this. The efficacy of the games was tested by demonstrating improvement in lower limb function by completion time in the 10 Meters Walk Test score (*Palacios-Navarro, García-Magariño & Ramos-Lorente, 2015*); and upper limb function by evaluating performance using Jamar handgrip dynamometer, Box and Blocks Test and Purdue pegboard test (*Oña et al., 2018*). In this group then, SCMoCap offered real value to the treatment of patients by enabling the capturing of performance data not easily achievable by other means.

SCMoCap was also used in a number of studies to discriminate between Parkinson's and non-Parkinson's patients (*Galna et al., 2014*; *Rocha et al., 2014*; *Rocha et al., 2015*; *Eltoukhy et al., 2017*; *Butt et al., 2017*; *Lee, Sinclair & Jones, 2019*) using upper and lower limb kinematic measurements. Upper limb measurements mainly focussed on velocity, acceleration and frequency of the hand and wrist during activities such as supination and pronation; hand opening and closing; and finger tapping (*Galna et al., 2014*; *Butt et al., 2017*; *Lee, Sinclair & Jones, 2019*). Lower limb measurements mainly focussed on collecting acceleration, velocity, distance between legs, range of motion, stride duration and cadence during gait (*Galna et al., 2014*; *Rocha et al., 2014*; *Rocha et al., 2015*; *Eltoukhy et al., 2017*). *Butt et al. (2017)*, *Lee, Sinclair & Jones (2019)* compared these measurements against those obtained using the MDS-Unified Parkinson's Disease Rating Scale (*Goetz et al., 2008*), a clinical scale used to evaluate Parkinson's disease severity with items that focus on motor function. *Butt et al. (2017)* found difficulty recognising forearm supination and pronation, finger trajectories and tremors due to the high speed of the movements, resulting in

limited clinical association. The ability to detect bradykinesia, slowness of movement associated with Parkinson's, had strong agreement with clinical scores using measurements of wrist supination and pronation, and hand opening and closing movements; however, they found that finger tapping contributed little predictive power to the model due to inaccuracies with tracking using a Leap Motion (*Lee, Sinclair & Jones, 2019*). *Galna et al. (2014)* compared Kinect measurements against a 3D marker-based system during gait and upper limb reaching and grasping tasks. They found the Kinect to be capable of measuring temporal and gross spatial parameters; however, it lacked the precision to track minor spatial movements of the hands and fingers accurately.

In multiple sclerosis (MS), studies aimed to quantitatively assess functional decline by measuring gait and postural control. In this population, motion capture typically involves GAITRite (CIR Systems Inc. Clifton, NJ, USA), a pressure sensitive walkway that provides spatiotemporal markers of gait (*Sosnoff, Sandroff & Motl, 2012*), and marker-based systems (*Galea et al., 2017*). These methods are highly accurate (*Shanahan et al., 2018*), but they require a laboratory environment and a long setup time, and markers may obstruct natural gait. Several studies investigated the use of SCMoCap to quantify the Short Maximum Speed Walk (SMSW), a new proposed observer independent measure of detecting walking speed and sway using hip joint centre tracking (*Behrens et al., 2014*; *Behrens et al., 2016*; *Grobelny et al., 2017*). They compared SMSW against established observer based clinical measures of gait disability such as 25-foot walk (*Polman & Rudick, 2010*), where maximum walking speed is assessed over a distance of 25 ft by an observer with a stopwatch and is often used as part of the Multiple Sclerosis Functional Composite (MSFC), and Expanded Disability Status scale (*Kurtzke, 1955*), a scale used to determine how MS affects the individual by assessing function such as muscle weakness and balance. SMSW was found to be superior to T25FW in the most recent study as it can detect walking speed over one or two strides—making this a more suitable test for those with limited gait function. This is a promising application of SCMoCap, as these measures rely on large-scale planar movements and do not require the accurate measurement of small segment motions or axial rotation of segments. In this scenario, SCMoCap can effectively replace more cumbersome setups such as the walkway.

In stroke patients, inertial (*Mizuike, Ohgi & Morita, 2009*) and 3D optoelectronic systems (*Chen et al., 2021*) have been used to assess movement with high accuracy. SCMoCap studies involving people who have had a stroke mainly focused on collecting kinematics that occur during single outcome clinical assessments but are not typically measured, to provide additional kinematic information about performance that could be useful for treatment. The primary focus of these studies was gait (*Vernon, Paterson & Bower, 2015*) assessments such as 10 meter walk test (10 MWT) and Timed up and go (TUG); and upper limb assessments (*Ozturk et al., 2016*) such as Wolf Motor Function Test (WMFT), which scores motor ability during functional tasks (*Wolf et al., 2005*). (*Vernon, Paterson & Bower, 2015*) collected gait kinematics during a TUG such as range of trunk angle, step length, stride length, cadence and gait speed to predict outcomes on TUG and 10MWT performed by a clinician and found these measurements to accurately predict outcomes. *Ozturk et al. (2016)* collected upper limb kinematics such as range of motion, speed and joint trajectories

with a Kinect during a reaching task and could predict performance on clinical WMFT and discriminate between stroke and healthy controls. *Scano et al. (2014)* compared upper-limb kinematics collected from a Kinect v1 against a 6 Camera 3D marker-based system (SMART BTC, Italy) during a reaching against gravity task. This was one of the few studies to report joint angle error of measurement in degrees; where shoulder elevation angle had a reported difference of 3.32 degrees $\pm$ 2.80 degrees and elbow flexion-extension angle had a reported difference of 5.60 degrees $\pm$ 6.35 degrees. Both errors of measurement were reported to be clinically acceptable.

An epilepsy diagnosis depends on electroencephalogram (EEG) readings and characteristics of seizure induced movements. Inertial sensors (*Nijsen et al., 2005*) and marker-based methods (*Cunha et al., 2012*) have been used to analyse seizure movements in Epilepsy Monitoring Units, with the advantage of being a quantitative approach to studying movements of interest that were traditionally only assessed by visual inspection from a clinician. However, several impracticalities have resulted from this method: mainly, the difficulty in attaching sensors and markers in a way that they do not detach during seizures, resulting in these methods being unsuitable for violent seizures. One Kinect per patient was used to count and discriminate between seizure types in an Epilepsy Monitoring Unit (EMU) setting in one study (*Rémi et al., 2011*; *Pereira Choupina et al., 2018*). Kinematics collected during a seizure included trunk speed, wrist speed and trajectories of trunk center and wrists. The seizure movements demonstrated high correlation with marker-based measurements (*Cunha et al., 2016*) and seizure classifications were validated by two clinicians (*Pereira Choupina et al., 2018*).

The benefit of the SCMoCap approach in adults with neurological injury compared to marker-based systems is reduced setup time, and the avoidance of potentially intrusive markers. The main technical limitation reported was accuracy ($n = 5$), specifically with high frequency movements, such as tremors (*Chen et al., 2016*; *Butt et al., 2017*); forearm supination and pronation (*Butt et al., 2017*); minor movements, such as finger tapping (*Lee, Sinclair & Jones, 2019*) or hand clasping (*Galna et al., 2014*); low resolution of video in a study using only an RBG camera (*Rémi et al., 2011*); and general accuracy of sensor, even with the higher resolution version of the Kinect (*Nakamura et al., 2019*). Two studies reported having issues with getting patients to perform the correct movements in front of the camera (*Butt et al., 2017*; *Grunert, Krause & Feig, 2019*), but this is of course a limitation not unique to SCMoCap technology.

## Hereditary/genetic neuromuscular disorders

Hereditary disorders occur as a result of altered genes which can affect several signalling pathways in the body (*Mirkin, 2006*). These genes can be transmitted between generations and, depending on the pathway affected, result in changes to a range of body systems such as skeletal muscle and peripheral nerves which can affect movement (*Conceição et al., 2016*; *Lim, Nguyen & Yokota, 2020*).

SCMoCap in this group was utilised with the intention of monitoring disease progression and planning rehabilitative treatment. All studies used cameras that include depth sensors,

three studies used a Kinect v2 (*Carmo Vilas-Boas et al., 2019*; *Vilas-Boas et al., 2019*; *Vilas-Boas et al., 2020*) and two studies used a Kinect v1 (*Han et al., 2015*; *Han, Bie & Nicorici, 2016*). Depth sensors were utilised to be comparable to 3D marker-based systems and for studies involving understanding functional workspace (*Han et al., 2015*; *Han, Bie & Nicorici, 2016*). A 3D representation of the workspace was required to better track and understand the subject's movement trajectories within the workspace. Diseases in this group are rare and consequently there is limited kinematic data available on their progression; therefore, SCMoCAP is invaluable in the effort to understanding these diseases as it has the potential to facilitate practical and more frequent monitoring such as in the home of patients.

Transthyretin-Related Familial Amyloid Polyneuropathy (TTR-FAP) is an adult-onset, neurodegenerative disease causing muscular weakness and other systemic issues. Movement analysis in this group aims to support diagnosis and prediction of future impairment. Studies using the Kinect v2 aimed to differentiate asymptomatic, symptomatic and control groups by measuring spatiotemporal and kinematic parameters during gait (*Carmo Vilas-Boas et al., 2019*; *Vilas-Boas et al., 2019*). Spatiotemporal measurements of gait calculated included stride and step duration; step and stride length; step and gait speed. Kinematic parameters during gait included joint angles for the spine, elbow, knee and hip. They compared these to marker-based systems and found higher agreement for spatiotemporal features than kinematic. Most spatiotemporal parameters were found to be clinically relevant; however, out of the kinematics studied, only minimum elbow angle was found to be useful for gait assessment.

Limited upper extremity functional assessment methods exist for assessing patients with facioscapulohumeral muscular dystrophy (FSHD) and Duchenne muscular dystrophy (DMD). FSHD evaluation scale (*Lamperti et al., 2010*) involves scores provided by an observer, that aim to assess upper extremity function and are used to track muscle weakness and functional decline within disease progression. For DMD, Performance of Upper Limb assessment (PUL) (*Mayhew et al., 2013*) aims to track the progression in muscle weakness and decline of function. FSHD (*Han et al., 2015*) and DMD (*Han, Bie & Nicorici, 2016*) studies using SCMoCap aimed to compare reachable workspace as a measure against existing clinical evaluations of upper limb function, to better understand disease progression and monitor recent therapeutic interventions. The reachable workspace was calculated using parameters including movement trajectories and range of motion. FSHD scores were compared against a SCMoCap approach using a Kinect and were found to have high agreement. Reachable workspace showed strong correlation with PUL evaluation of DMD. FSHD and DMD studies reported that the reachable workspace outcome measure can differentiate upper extremity function levels with adequate sensitivity.

The main limitation reported by almost all studies in this group ($n = 4$) (*Han, Bie & Nicorici, 2016*; *Vilas-Boas et al., 2019*; *Vilas-Boas et al., 2020*) was small sample size. This was further associated with the rarity of the diseases, making recruitment of subjects for various disease stages difficult.

## Frailty

Frailty (*Xue, 2011*; *Clegg et al., 2013*) is a disorder associated with increased vulnerability, stemming from a decline in multiple body systems. This is associated with poorer health outcomes such as higher falls risk, hospitalisation and mortality and is usually classified according to the presence of phenotypes *e.g.*, slower walking speeds, low physical activity or unintentional weight loss. Frailty is distinct from normal processes associated with healthy ageing; however, some phenotypes used in the diagnosis of frailty may be present in some older adults.

The aim of most studies using SCMoCap in this group was to improve fall risk assessment. Studies measured at-home gait parameters: one study estimated Timed up and go time (TUG) (*Stone & Skubic, 2013*) which is used as part of a typical fall risk assessment; another used gait data collected to detect falls in the home (*Stone & Skubic, 2015*). The Kinect was used as a practical method to collect gait sequences in all studies. Other technologies used for fall detection include floor vibration (*Alwan et al., 2006*), passive infrared (*Sixsmith, Johnson & Whatmore, 2005*) and acoustic (*Zigel, Litvak & Gannot, 2009*) sensors. These methods provide mostly spatiotemporal data, whereas the Kinect has the ability to measure further information such as body segment kinematics preceding a fall, that may give insight into the causes of falls and lead to improved methods of falls prevention. Depth information from the Kinect was also utilised as it is robust to variation in ambient lighting and to understand the 3D space in which the fall takes place.

Spatiotemporal parameters of gait were mainly collected, such as speed, stride time and length, and height, over a period of up to 16 months. The SCMoCap studies evaluated the technology using the clinical assessment of TUG against SCMoCap-estimated TUG from in home gait speed (*Stone & Skubic, 2013*), statistical validation against state-of-the-art fall detection algorithms (*Stone & Skubic, 2015*) and a comparison against hand labelled data (*Stone & Passive, 2012*). In home gait speed was found to be accurate at predicting TUG performance. All studies reported success in capturing the majority of natural gait sequences and falls. *Stone & Skubic (2015)* reported technical limitations with occlusion and issues with limited field of view, requiring falls to be in the view of the sensor which made the capture of some gait sequences not possible. This emphasises the importance of camera positioning in such studies, and suggests that the size of the space being monitored needs to be carefully considered.

## Orthopaedic or musculoskeletal

The classification (*Punnett & Wegman, 2004*) of musculoskeletal or orthopaedic disorders encompasses a broad range of conditions, such as arthritis or tendinitis, with varied disease mechanisms and causes *e.g.*, acute trauma, overuse trauma, degeneration or inflammation. Whilst the skeletal and muscular systems are predominantly affected *e.g.*, joints, muscles, and tendons, it can also include peripheral nerves and associated vascular structures. These conditions can impair movement and can be managed using surgical and non-surgical interventions. SCMoCap in this group was mainly carried out to compare the accuracy of the Kinect with clinical outcome measures. All studies in this group used the Kinect to

provide a 3D understanding of movements produced by multiaxial joints such as the hip and shoulder.

One single case study aimed to monitor changes in gait before and after total hip replacement surgery (*Dolatabadi, Taati & Mihailidis, 2014*) with the intention of understanding balance issues after surgery. Outcome measures assessing balance after surgery are typically in the form of self-report which does not always align with clinical assessment (*Gandhi et al., 2009*). To achieve this spatiotemporal and kinematic parameters were calculated using a Kinect. Spatiotemporal measurements collected during gait included step length, stance time and cadence. kinematics collected during a sit to stand exercise included hip angular velocities and center of mass. Results obtained were statistically compared against the subjects own self-assessment during rehabilitation and were found to agree well.

Another tested the agreement of Kinect measurements against clinical goniometer readings of upper limb range of motion in patients with adhesive capsulitis of the shoulder (AC); during flexion, abduction and external rotation (*Lee, Yoon & Chung, 2015*). AC is typically diagnosed based on shoulder range of motion measurements usually with a goniometer. Goniometer measurements are standard in clinical practice; however, results can differ between observers (*Riddle, Rothstein & Lamb, 1987*). The Kinect was demonstrated by this study to be a clinically viable tool for assessing active and passive shoulder range of motion and diagnosing AC, given the excellent agreement with goniometer readings. Inertial (*Mullaney et al., 2010*) and 3D marker-based (*Ropars et al., 2015*) methods have been used to assess shoulder range of motion in the past with excellent accuracy; however, these methods require setup and do not compare with the practicality of the Kinect.

## CONCLUSIONS

SCMoCap helps to facilitate motion analysis in situations where it would otherwise not be possible, such as at-home rehabilitation for persons with Parkinson's (*Palacios-Navarro, García-Magariño & Ramos-Lorente, 2015*; *Oña et al., 2018*; *Grunert, Krause & Feig, 2019*), where rehabilitation feedback could be personalised to the individual. Given that limited accuracy as a result of camera occlusion or difficulties with recognising movement characteristics was reported by many studies ($n = 10$) (*Adde et al., 2010*; *Galna et al., 2014*; *Vernon, Paterson & Bower, 2015*; *Chen et al., 2016*; *Butt et al., 2017*; *Støen et al., 2017*; *Latorre et al., 2018*; *Lee, Sinclair & Jones, 2019*; *Nakamura et al., 2019*; *Tsuji, Nakashima & Hayashi, 2020*), it would be inappropriate to use a SCMoCap approach instead of a marker-based approach in situations where detailed 3D kinematics or the granularity of kinematic measurements is paramount, such as surgical planning.

This technology could also be considered when a marker-based setup would cause distress to a patient, for example, in the analysis of seizures (*Rémi et al., 2011*; *Pereira Choupina et al., 2018*) or can be difficult or inconvenient, for example in young children with cerebral palsy. Clinical spatiotemporal gait parameters measured in this group presented high agreement with Gross Motor Function Classification System (GMFCS)

and Gait Deviation Index (GDI) scores (*Kidziński et al., 2020*) evaluated by clinicians—introducing the possibility for a markerless cerebral palsy gait assessment. High agreement between General Movement Assessment (GMA) and a SCMoCap approach in the early identification of cerebral palsy was demonstrated across multiple studies (*Adde et al., 2010*; *Orlandi et al., 2015*; *Orlandi et al., 2018*; *Støen et al., 2017*; *Marchi, Hakala & Knight, 2019*; *Schroeder, Hesse & Weinberger, 2020*; *Tsuji, Nakashima & Hayashi, 2020*; *Reich et al., 2021*; *Groos et al., 2022*); this leads to the possibility of GMA being performed by parents or guardians entirely at home, or as a precursor to a clinical GMA.

Generally, measured movements were linked to easily identifiable events, or were indicative of disease conditions, *e.g.*, seizures in epilepsy, falls in frailty, or tremor in Parkinson's. The most promising results involved infants with neurological injury—this possibly stems from the fact that all movements of interest can be captured in a single position (supine) and so overcomes challenges associated with multi-planar movements where landmark identification is occluded by other body parts as a result of using a single camera approach.

In most studies, repeatability was assessed using correlation coefficients and other statistical measures, from which the ability to derive absolute error of measurement is not possible. Few studies ($n = 5$) compared against a marker-based approach or other unit appropriate baseline and reported error of measurement in units consistent with the measurement *e.g.*, degrees for joint angles (*Galna et al., 2014*; *Scano et al., 2014*; *Carmo Vilas-Boas et al., 2019*; *Vilas-Boas et al., 2019*; *Kidziński et al., 2020*). Furthermore, only five studies (*Stone & Skubic, 2015*; *Rahmati et al., 2016*; *Nakamura et al., 2019*; *Kidziński et al., 2020*; *Tsuji, Nakashima & Hayashi, 2020*) shared an algorithm and datasets, and only one study shared code (*Kidziński et al., 2020*). This makes the repeatability of methods onerous and hinders clinical implementation. The sensitive nature of the data used in most studies explains the necessity for privacy; however, a more abstract approach to data sharing such as the sharing of key point data would still be invaluable and aid in maintaining the privacy of patients. There is significant variability in the way in which SCMoCap studies are reported, in terms of the features measured, and methods selected to evaluate the systems. As a result, there is no universal metric or outcome that can be used for comparison between studies. If SCMoCap technology is to be implemented in clinical practice, future work should look to develop a reference framework which would facilitate comparison of studies in this field.

The portability of SCMoCap systems compared to multi-camera is irrefutable, however, within SCMoCap systems some methods allow for more freedom in data acquisition than others. Devices such as the Kinect and Leap Motion require a computer nearby for data collection. On the other hand, software-based solutions such as OpenPose or DeepLabCut can utilise input that has been collected earlier from any single camera system, meaning data collection is less restricted and may be more feasible outdoors. It is worth noting that most software solutions investigated such as OpenPose and DeepLabCut, are still computationally expensive to run and would most likely not run in real time on most home computers. MediaPipe (*Bazarevsky et al., 2020*) aims to alleviate this issue by making pose estimation accessible in a web browser. No studies in this review were found to use MediaPipe with patient populations. In addition to portability considerations, most

 

software solutions used require a level of programming expertise to run, involving tasks such as preparing input data and evaluating the model.

Depth sensors, while being able to provide useful 3D data inherently without 2D to 3D projection (*Zhang, Zhan & Chang, 2021*), suffer from lower framerates, light interference and are limited to a smaller capture space (*Clark et al., 2019*) than standard video cameras. This limits their potential in large areas and outdoors; however, for telerehabilitation purposes they remain a viable solution for indoor home use in most cases. Furthermore, many modern smartphones are equipped with depth sensors so the issue of depth sensors only being available to specialised devices that require a connection to a computer for data collection and processing may be mitigated with time.

Limitations involving accuracy due to the device, such as field of view, resolution, and frame rate, will be naturally ameliorated with time as these improvements become more available and affordable in consumer devices. Other limitations such as occlusion (*Lee, Sinclair & Jones, 2019*) and quantifying the quality of movements and gestures (*Orlandi et al., 2015*; *Tsuji, Nakashima & Hayashi, 2020*) invite more research as they require a deeper understanding of the movement performed than just the kinematic measurements detectable by the camera—these issues transcend even the gold standard multi-camera 3D marker-based approach for this reason.

Overall, clinical implementation of this technology in most groups, aside from neurological injuries in children, is hindered by limitations both in study design and technology. Most studies are proof of concepts with small sample sizes. Errors of measurement are also not reported appropriately, such as reporting error of joint angles in degrees, so we cannot know how the errors of measurement will influence clinical decision making—which will be required for clinical implementation. Furthermore, the tolerance for the errors of measurement is likely to be a consequence of the granularity required by the clinical intervention. Nevertheless, SCMoCap is a promising technology that has the potential to be an invaluable tool in the patient pathway for movement disorders, taking motion capture outside of the laboratory and into patients' homes. Further research should focus on improving tracking accuracy of movements that are out of plane relative to the camera orientation or affected by occlusion, such as measuring supination and pronation of the forearm with a camera oriented in the frontal plane. A hypothetical approach to solving this problem may be to understand physiologically what is out of sight of the sensor through inverse kinematics by using musculoskeletal models (*Chadwick et al., 2014*) in tandem with high accuracy machine learning skeletal tracking models (*Cao et al., 2019*).

### Funding

This work was funded by a University of Aberdeen Elphinstone PhD scholarship. The funders had no role in study design, data collection and analysis, decision to publish, or preparation of the manuscript.

## Grant Disclosures

The following grant information was disclosed by the authors:
University of Aberdeen Elphinstone PhD scholarship.

## Competing Interests

The authors declare there are no competing interests.

## Author Contributions

- Bradley Scott conceived and designed the experiments, performed the experiments, analyzed the data, prepared figures and/or tables, authored or reviewed drafts of the article, and approved the final draft.
- Martin Seyres conceived and designed the experiments, performed the experiments, analyzed the data, authored or reviewed drafts of the article, and approved the final draft.
- Fraser Philp conceived and designed the experiments, analyzed the data, authored or reviewed drafts of the article, and approved the final draft.
- Edward K. Chadwick analyzed the data, authored or reviewed drafts of the article, and approved the final draft.
- Dimitra Blana analyzed the data, authored or reviewed drafts of the article, and approved the final draft.

## Data Availability

   The raw data is available in the Supplementary File.

## Supplemental Information

Supplemental information for this article can be found online at http://dx.doi.org/10.7717/peerj.13517#supplemental-information.

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
