# Peer review of "Healthcare applications of single camera markerless motion capture: a scoping review"

_PeerJ, doi:10.7717/peerj.13517_

## Round 0.1 · original submission · Minor Revisions

The reviewers have provided very positive reviews and suggested a few minor edits for clarity.

·

Basic reporting

Overall the paper is well written and easy to follow, with a clear message. Figures are also very clear. A couple of comments:

It would be helpful to note early on that “single camera motion capture systems” refers to both hardware-based solutions (e.g., Kinect, Leap Motion), as well as video-based offline solutions (e.g., OpenPose, DeepLabCut).

Line 245 states, regarding neurological injuries in adults: “in all cases voluntary movement is affected.” Is this a general statement about neurological injury? Or about the studies being included?

Experimental design

Was there a reason for not including psychiatric patient groups (e.g., autism, schizophrenia)? Or were they included under one of the other headings?

Validity of the findings

The authors comment on single camera motion capture being a much more accessible solution to kinematic analysis than more complex, expensive systems. While I very much agree, I wonder if the authors can comment on how ‘accessible’ the different single camera systems may be in different circumstances. For example, Kinect V2 requires a computer set up to run the Kinect, making this at least somewhat more cumbersome than video-based methods that only require a camera. On the other hand, video-based methods may require more expertise and/or computing power to actually process the data. This is particularly true of DeepLabCut, and to a less extent also OpenPose. Newer methods, such as MediaPipe doesn’t have this restriction as much, but have other limitations in its processing possibilities.

It would be helpful to indicate which system of motion capture was used in each study. For example, “The ability to detect bradykinesia, slowness of movement associated 296 with Parkinson’s, had strong agreement with clinical scores using measurements of wrist 297 supination and pronation, and hand opening and closing movements; however, they found that 298 finger tapping contributed little predictive power to the model due to inaccuracies with tracking 299 (Lee, Sinclair & Jones, 2019)”. Knowing which systems succeeded in which tasks and populations (which is indicated in some paragraphs, but not others) would be quite useful information to include throughout the results, in my opinion.

As an extension of one of my previous questions, it may be useful to comment in the discussion on which method(s) may be most promising (e.g., based on what they can do, but also hardware requirements, etc), perhaps for different patient groups, or in different data acquisition contexts. If this is possible with this type of review.

Additional comments

No comment

·

Basic reporting

The English is clear and professional and absolutely suitable to be read by an international audience.
The literature references show a detailed and in-depth knowledge of the clinical applications of markerless systems.
The structure of the article is consistent with PeerJ standards. In particular, there is an abstract divided into four sections: Background, Survey methodology, Results and Conclusions. The introduction describes correctly the field and its importance by clarifying which is the aim of this review (e.g. from lines 107 to 109 “this review provides insight into the healthcare application of SCMoCap to inform clinicians…” or from lines 111 to 114 “a scoping review was therefore conducted with the aim of investigations: which SCMoCap systems …”). In my opinion, the introduction should cite other previous reviews in the field of markerless analysis. The survey methodology is well-written since inclusion and exclusion criteria are reported.
To the best of my knowledge, the field of markerless analysis has not been reviewed recently and for this reason I can affirm that it is necessary a review that summarizes and clarifies all the information provided in the literature in this period.

Experimental design

In my opinion, the investigation is sufficiently rigorous and reflects the aims and scope of the journal. The methods are described with sufficient details. In particular, the search strategy, the inclusion and exclusion criteria are reported and there is also a figure which represents the flow chart of the review with which another investigator can replicate the same investigation.
The unbiased coverage of the subject is guaranteed by the presence of two investigators who have selected and read all the articles.
The sources of the selected articles are adequately cited.
The best point of this review is the organization of the paragraphs because the field of applicability of the markerless systems is very wide because it could include neurological injuries in Children and Adults, Neuromuscular Disorders and Frailty. The presence of these paragraphs makes the reading of the results clearer and more fluent. In addition, each paragraph contains a brief introduction about the selected disease for the readers who are not confident with it.

Validity of the findings

The conclusions are well written since they are connected with the research questions reported in the introduction.
In particular, both advantages and disadvantages of markerless systems are reported. I agree with the sentence reported from lines 499 to 501 since in my opinion markerless systems are suitable especially for screening purposes.
In the conclusions, unresolved questions and future directions are identified. In particular from the lines 558 to 563 the article reported the improving of tracking accuracy in presence of occlusions or of movements out of plane relative to the camera orientation as a future research.

---

## Round 0.2 · accepted · Accept

Thank you for your excellent submission. This will be of interest to many wanting to apply motion capture in clinical areas but lacking the resources or practical means to employ traditional instruments and methods.